# Labor Market Segmentation and Immigrant Competition: A Quantal Response Statistical Equilibrium Analysis

**DOI:** 10.3390/e22070742

**Published:** 2020-07-05

**Authors:** Noé M. Wiener

**Affiliations:** Department of Economics, University of Massachusets, Amherst, MA 01002, USA; nwiener@umass.edu; Tel.: +1-413-545-6357

**Keywords:** labor market competition, wage inequality, statistical equilibrium, immigration

## Abstract

Competition between and within groups of workers takes place in labor markets that are segmented along various, often unobservable dimensions. This paper proposes a measure of the intensity of competition in labor markets on the basis of limited data. The maximum entropy principle is used to make inferences about the unobserved mobility decisions of workers in US household data. The quantal response statistical equilibrium class of models can be seen to give robust microfoundations to the persistent patterns of wage inequality. An application to labor market competition between native and foreign-born workers in the United States shows that this class of models captures a substantial proportion of the informational content of observed wage distributions.

## 1. Introduction

Labor competition plays a central role in the determination of wages in labor markets. On the one hand, wage differentials orient the behavior of workers in terms of job changes, geographical migration, educational investments and other decisions regarding their labor supply. On the other hand, the cumulative but unintended effect of these individual behaviors significantly shapes wages in the different labor markets. An ongoing concern of political economists has been the extent to which different groups participate in the process of labor market competition [1,2,3]. In particular, how are wages of immigrants and natives shaped by their interaction in the labor market? Are native workers competing for the same positions as immigrants, or are they mostly located in different segments of the labor market [4,5]?

Social researchers are generally limited in their ability to study the interaction between worker behaviors and wage outcomes by the coarse nature of their data. Large-scale social surveys are designed to capture the current composition and socioeconomic status of the population, and researchers need to draw inferences about the unobserved aspects of behavior that are relevant to their work. In the study of labor market dynamics, interest often centers on the extent to which different groups of workers are competing with each other. This depends on factors such as informational limitations, costs of job change and the degree of market segmentation, all of which are hard to define and measure.

This paper applies a parsimonious class of social interaction models, the quantal response statistical equilibrium (QRSE) model developed in [6], to the case of labor markets. As shown below, the model is inspired by what may be called the Smithian conception of competition. The QRSE class of models not only provide a very good fit to the wage data, drawn from the US Census, but also allow inferences about the competitive behavior of workers. This is interesting for social scientists who generally have to deal with incomplete and noisy data, particularly with regard to behavioral information. The models proposed here rely on the maximum entropy principle to make such inferences about the relationship between the observed wages and the unobserved behaviors of individuals. From the inferred joint distribution of wages and actions, we can extract readily interpretable measures of the intensity of competition in labor markets.

Beyond the application of the QRSE models to a new type of market, this paper also extends the framework by considering the interaction of multiple types of agents that differ in their behaviors. It is argued that this heterogeneous agent version of the QRSE model provides a helpful tool for analyzing competition among social groups in markets characterized by extensive segmentation. This paper presents estimates of the competitive reach of immigrant and native workers in the US over the past 50 years on the basis of census data.

In the first part of this paper, we discuss the incomplete data problem in population surveys, and the difficulties it poses for measures of labor competition. The maximum entropy approach is proposed as a solution. The QRSE class of models of labor market competition is then outlined, and extended to make minimally biased inferences about the unobserved behaviors of different groups. In the second part, we show how our model class can be applied to Census data in the US by broad racial and ethnic groups. Despite its simplicity, the QRSE model class can robustly account for observed differences in wage distributions between immigrants and native-born white, Black and Hispanic workers. Finally, the relationship with the broader literature on immigration in labor markets and wage inequality is discussed.

## 2. Labor Market Competition and the Incomplete Data Problem

Labor force surveys and population censuses have been the basis of most research in empirical immigration economics. Census data are primarily designed to estimate the sizes of different subpopulations, including immigrants from different regions of origin, and a small set of their socioeconomic characteristics. While this allows for a stratified analysis of national labor markets, notably by skill level [4] and region [7], the information necessary to measure labor market competition unambiguously is missing.

Important aspects of worker behavior are unobservable through social surveys, posing difficulties for the measurement of labor market segmentation. In an ideal world, labor economists would have access to detailed individual employment histories and associated information about wage levels and full occupational characteristics for the entire population. This would allow us to observe most of the competitive behaviors shaping the wage distribution; in particular, any actions that alter the labor supply in particular market segments. In practical research, our data are of a much more limited kind. No matter how detailed the survey, there will still be considerable ambiguity in defining the exact contours of a labor market segment. There is simply no satisfactory way of accounting for the various spatial, occupational and socio-cultural dimensions that prevent labor mobility.

As an exemplary case, consider the census data used in this paper. The census has very limited information on behavioral aspects of labor competition such as job changes and prior employment. More detailed longitudinal surveys on career trajectories exist, but such studies offer much smaller sample sizes. None of these data sources offer the type of fine-grained occupational information necessary to define labor market segments in any unambiguous sense. It is therefore desirable to find ways of making use of the type of information available in population surveys and censuses.

The general class of models developed in [6], and further elaborated in [8,9], offers a transparent and minimally biased measure of competition in this type of incomplete data setting. This approach is presented here as a heuristic tool of statistical inference, and an outline for further conceptual developments in the political economy of labor. The methodological point-of-view underlying the argument is sufficiently different from standard economic practice to warrant a brief recapitulation.

As discussed above, survey data represent a snapshot of labor market *outcomes* at a given point in time, without detailed information on individual trajectories. In such circumstances, theoretical considerations are usually drawn upon to complement the observed data. While economic theory can provide some broad guidelines about causal relationships in social interaction settings, there is very little consensus on the specific micro-kinetic laws governing behavior [10]. Rather than proposing a specific set of behavioral rules which are then exposed to falsification along the lines of [11], the approach followed here attempted to identify the least restrictive, most general *class of models* still consistent with the observed data. Rather than claiming to have identified the uniquely correct model of labor market competition, the goal was rather to provide intermediate results that help focus future research.

That goal was achieved by deriving a set of *probabilistic* associations between observed wage outcomes and the unobserved competitive behavior, following a broadly Smithian vision of competition [12]. A range of specific micro-kinetic models of worker mobility and competition will be compatible with the joint probability distribution proposed here. The model class is defined by a set of constraints on moments of the joint distribution, which embody the common Smithian theme of competition as a negative feedback mechanism [13].

### A Maximum Entropy Solution

The problem of inferring information about the unobservable dimensions of labor market competition on the basis of incomplete data is approached using an axiomatically well-founded information-theoretical tool. The maximum entropy (MaxEnt) approach, developed in a series of contributions by Edwin Jaynes [14,15], can be summarized most immediately as a heuristic for choosing a probabilistic model that incorporates any prior information about a system under study, while imposing no additional hidden or extraneous assumptions. In economic systems, this prior knowledge includes, in particular, information about the structure of incentives and constraints faced by the interacting agents [8].

Entropy is a measure of uncertainty, or equivalently of the expected amount of information conveyed by observing the outcome of an experiment. Intuitively, the surprise at observing an outcome *k* should depend negatively on its probability of occurrence $p_{k}$. A rare event should surprise an observer more than a common event. The negative logarithm of the probability of observing the event, $-log(p_{k})$, was shown by Shannon to be a consistent measure of the amount of information in outcome k. The base of the logarithm defines the unit for the measure of uncertainty, such as *bits* for a base of 2. As an example, consider the experiment of flipping a biased coin with a prior probability of coming up with heads $p_{h}=0.9$. The amount of information conveyed by an outcome “tails” is $-log_{2}\left(0.1\right)\approx 3.3$ bits.

On the other hand, the *expected* surprise value of a rare event before the outcome is known should be quite low. The ex-ante uncertainty about the outcome of flipping the biased coin is much lower than the uncertainty for a fair coin. Reference [15] has shown that the unique measure of expected surprisal (or average observer uncertainty) that respects a small set of axioms is Shannon’s entropy:(1)$$S\left(p\right)=-\sum_{k}p_{k}\;log\left(p_{k}\right)$$

The goal of inferring a probability distribution over possible outcomes without imposing any hidden assumptions can be seen to be tied to maximizing the uncertainty (Equation 1) of the candidate distribution. The MaxEnt principle directs the researcher to express her theoretical expectations in the form of constraints on moments of the probability distribution, and then select the least committal, most uncertain distribution that still satisfies the constraints.

Conversely, [15] has shown that given a set of constraints, realized distributions in systems with many degrees of freedom are overwhelmingly more likely to be near the entropy-maximizing distribution. This result has generated a research agenda of identifying distributions of social outcomes that show some structure and stability over time and interpreting them *as MaxEnt distributions*. The subsequent step is to deduce the set of constraints subject to which these candidate distributions would be entropy maximizing.

These constraints in the MaxEnt program constitute in effect the class of models that are consistent with the evidence. While it may be possible to devise more elaborate micro-level models, they must imply the same set of constraints on the moments of the joint probability distribution. Since the evidence does not support further distinctions among potential micro-level models, parsimony would suggest reporting the constraints directly. In economics, recent work has applied this particular approach to the study of profit rates [16], wages [17], the growth rates of labor and capital productivity [18] and Tobin’s q [19]. See [20] for a recent review of maximum entropy applications in economics.

## 3. Smithian Class of Labor Competition Models

The conceptualization and role of labor competition from the Smithian and classical points-of-view are active research areas [21]. Here, we take the theory to imply that workers respond to conditions of employment by entering or leaving particular labor market segments. In the process, entering workers will tend to place downward pressure on niche conditions and vice-versa for leaving workers, thereby reducing the differentials that gave rise to the mobility in the first place. Conditions of employment include a range of both observable and unobservable features and amenities, broadly related to the ratio of labor effort required in the position to the returns from employment. At first approximation, we can argue that (observable) wage rates serve as indices of these overall conditions of employment, given that labor mobility would tend to make returns proportional to labor effort in the long-run. If this is the case, we would expect the individual and uncoordinated entry/exit behavior of workers to lead to an organized and predictable distribution of wage rates as an unintended aggregate outcome.

From the information-theoretic point of view, the mechanism of competition conveys information about wage outcomes by preventing them from straying too far from a common center of gravity. The amount of uncertainty removed by the competitive process is a measure of its intensity. If a single wage prevailed across the entire labor market, this model would suggest that the market was perfectly competitive in the sense of standard economic theory. On the opposite end of the spectrum, wages would be completely disorganized. A rational observer would, when asked to predict the wages of a particular worker, merely pick any positive number at random. In this situation, Smithian competition provides no information about wage outcomes because it is inoperative. The Smithian class of models would interpret this situation to be one of complete segmentation, with no mobility between the labor market niches. The observed marginal wage distributions for the US economy are, unsurprisingly, between these extremes.

### 3.1. Quantal Response Statistical Equilibrium

We review here the logic of the quantal response equilibrium model proposed for the general case in [6] and discuss its application to labor markets. Let (w,a) be the state variables of the system, corresponding to wages (observed) and entry-exit behavior (unobserved). The quantal response statistical equilibrium in labor markets is the joint frequency distribution f(w,a) that has maximum entropy subject to two substantive constraints. These constraints represent a theory of competition that links the entry-exit behavior of agents to the wage outcomes.

#### 3.1.1. Quantal Response Behavior

Workers respond to differences in conditions of work and employment by entering or leaving particular labor market segments. It is common practice (e.g., [22,23]) to model decisions of this type by stipulating that workers will choose the strategy with the higher payoff *u*. As argued above, wages may be viewed as suitable indices of the payoffs of different lines of employment given their regulation by the mobility of labor. We therefore consider payoffs to be a function of segmental wages and the action taken by the worker, u(w,a). In the absence of more detailed knowledge about the problem, it is convenient to assume a linear and symmetrical payoff function where u(w,enter)=(w−μ) and u(w,exit)=(μ−w). The parameter μ here represents a reference value for the wage.

Without further modification, the assumption of payoff maximization could fit no real-world data in which dispersion is the norm. The standard approach is to assume that deviations from the predicted behavior reflect unobserved heterogeneity or cognitive mistakes that can be captured by adding random errors to the payoff, where the particular distribution of errors is chosen for computational convenience or by reference to a law of large numbers. A more psychologically convincing and parsimonious account for the deviations in behavior is suggested by the rational inattention framework [24,25]. In this perspective, the decision-making process is conceived of as a communication channel in the sense of information theory. The typical agent receives signals such as the wage rate through a channel with unknown but finite capacity, a constraint which is interpreted as the utility cost of processing the signal. Both physiological limitations and efficiency considerations lead to the prediction that only limited resources will be allocated to information processing. This assumption is enough to predict a behavioral response that corresponds to the well known logit or quantal-response pattern.

Reference [6] propose a straightforward formalization of bandwidth-constrained behavior. Formally, the typical agent chooses the mixed strategy f(a|w) that maximizes her expected payoff, subject to a lower bound on the entropy of the strategy:(2)$$\begin{array}{ccc} \; & \max_{\left\{f(a|w)\geq 0\right\}} & \sum_{a}f\left(a|w\right)u\left(w,a\right)\\ \; & s.t. & \sum_{a}f\left(a|w\right)=1\\ \; & \; & -\sum_{a}f\left(a|w\right)log\left(f(a|w)\right)\geq S_{min}. \end{array}$$

The Lagrangian associated with this payoff maximization problem takes the form:(3)$$L=\sum_{a}f\left(a|w\right)u\left(w,a\right)-\lambda \left(\sum_{a}f\left(a|w\right)-1\right)+T\left(\sum_{a}f\left(a|w\right)log\left(f(a|w)\right)-S_{min}\right)$$

By solving the optimization problem using the Lagrangian method, one obtains predicted mixed strategies of the form $f\left(a|w\right)\propto e^{\frac{u(a,w)}{T}}$. Given the linear payoff structure and binary action variable assumed here, the conditional action frequencies are:(4)$$\begin{array}{cccc} f(enter|w) & =\frac{e^{\frac{w-\mu }{T}}}{e^{\frac{w-\mu }{T}}+e^{\frac{\mu -w}{T}}} & =\frac{1}{1+e^{\frac{-2(w-\mu )}{T}}} & \\ f(exit|w) & =\frac{e^{\frac{\mu -w}{T}}}{e^{\frac{w-\mu }{T}}+e^{\frac{\mu -w}{T}}} & =\frac{1}{1+e^{\frac{2(w-\mu )}{T}}} & =1-f(enter|w) \end{array}$$

This derivation of the familiar quantal response behavior helps illustrate the extreme assumptions embedded in expected utility theory. In the absence of a minimum entropy constraint, the response of the typical agent would be concentrated at the payoff maximizing strategy. As T→0, the predicted action frequencies (Equation 4) approximate Heaviside step-functions that imply immediate switching from exit to enter as w>μ. The general case is clearly one where there are limits to the information processing capabilities of the agents that introduce a non-zero lower bound to the entropy of the mixed strategy $S_{min}>0$.

While the payoff maximization problem (Equation 2) is a more intuitive starting point for social scientists familiar with the expected utility approach, it is important to note its mathematically dual representation as an entropy maximization problem. The dual entropy maximization program is:$$\begin{array}{ccc} \; & \max_{\left\{f(a|w)\geq 0\right\}} & -\sum_{a}f\left(a|w\right)log\left(f(a|w)\right)\\ \; & s.t. & \sum_{a}f\left(a|w\right)=1\\ \; & \; & \sum_{a}f\left(a|w\right)u\left(w,a\right)=\bar{u}\left(w,a\right). \end{array}$$

An observer whose only knowledge about the decision-making process is that the expected payoff is finite, would similarly predict conditional action frequencies of the quantal response form (see, e.g., [26,27]). Both the payoff maximization problem subject to a minimum entropy constraint, and the entropy maximization problem subject to an expected payoff constraint, resolve themselves into the same Lagrangian and hence the same predicted quantal response distributions (Equation 4).

#### 3.1.2. Competitive Feedback

The tendential regulation of social outcomes by the uncoordinated behavior of economic agents is a core insight of political economy, famously epitomized by Adam Smith as the “invisible hand” of markets [12]. One relevant set of behaviors involves the entry and exit behavior of workers across labor market niches. As an unintended social consequence of their action, entrants tend to lower wages in their target niche (and vice-versa for leavers), thereby limiting the wage differentials that gave rise to the action in the first place. Adapting from [6], we may represent this feedback mechanism in a constraint on the difference between the expected wage of entrants and leavers:(5)f(enter)E(w|enter)−f(exit)E(w|exit)≤δ

As discussed above, if competition imposed no structure whatsoever on the system, the wages in advantageous labor market segments would tend to be infinitely greater than those in disadvantageous segments. The intuition behind (Equation 5) is that competition limits this divergence to some unknown but finite amount δ.

Institutional features of the labor market make it likely that the impact of competition is unevenly felt. Employment contracts differ systematically from one another, including in terms of the commitment involved. Employers conceive of positions with high turnover and tight control, and positions with internal career progression and greater autonomy. Considerations of efficiency, quality and power will lead to systematic associations between these job characteristics and their incentive schemes, including remuneration [28]. One possible adjustment to account for these institutional features in labor markets is to allow for the effects of entry and exit behavior to differ in magnitude. (Reference [29] proposed asymmetric feedback constraints in the context of liquidity in a stock market model.) This would allow for competition to have a potentially greater impact on low-wage sectors with highly flexible employment conditions, compared to a weaker impact of entry on higher-wage segments with more stable conditions.
(6)$$\mu -f\left(exit\right)E\left(w|exit\right)\leq \delta_{x}$$
(7)$$f\left(enter\right)E\left(w|enter\right)-\mu \leq \delta_{e}$$

We will present a comparison of empirical fits for the symmetric and asymmetric specifications below.

### 3.2. Labor Market Competition with Heterogeneous Workers

The standard QRSE model specifies the behavioral response of a “typical” or average worker. It is remarkable, then, that QRSE models predict inequality of outcomes despite such a parsimonious assumption. Reference [30] has highlighted how statistical equilibria generate horizontal inequality among identical agents, in contrast to the “equal treatment property” of general equilibrium models. While general equilibrium approaches trace observed inequalities back to (exogenous) differences in endowments and preferences of agents, statistical equilibrium models such as the QRSE emphasize market institutions and processes as endogenous sources of inequality.

Parsimony alone is an insufficient guide for model selection in the social sciences, particularly when social identity is expected to play a central role. Labor market competition is precisely an occasion wherein a more disaggregated analysis may be desirable. We represent differences between groups of workers in terms of their average *behavioral* responses to labor market opportunities, as captured by group-specific decision temperatures $T_{m}$ and reference wages $\mu_{m}$. However, native and immigrant workers both are part of the same labor market, even though concentrated in different market niches. Therefore, we keep the constraint that the expected wage of entrants, both immigrants and natives, exceeds that of leavers. This represents the notion that workers of all groups still compete in the same overall labor market, even if they differ in their access to different labor market segments and their willingness to accept low wage jobs. The assumption also appropriately reflects that we have insufficient information about specific patterns of cross-group competition.

Finally, we add a constraint on the proportion of agents of each type *m* (Equation 12). While the labor force participation of different social groups itself is endogenous to the functioning of labor markets, incorporating this feature into our model is beyond the scope of this paper. We therefore take the expected proportion of agents of each type to be equal to their observed proportion in our sample.

The maximum entropy program looks like:(8)$$\underset{\left(\{f(a,w,m)\geq 0\}\right)}{max}S\left(f\left(a,w,m\right)\right)$$
subject to
(9)$$\sum_{m}\sum_{a}\int f\left(a,w,m\right)\;dw=1$$
(10)f(enter)E(w|enter)−f(exit)E(w|exit)≤δ
(11)$$\begin{array}{cc} f(enter|w,m) & =\frac{1}{1+e^{\frac{-2(w-\mu_{m})}{T_{m}}}}\\ f(exit|w,m) & =1-f(enter|w,m)\;\forall \;(w) \end{array}$$
(12)$$f\left(m\right)=\frac{n_{m}}{\sum_{j}n_{j}}\;\forall \;\left(m\right)$$

After some manipulations (see Appendix A), we find that this maximization problem has a unique solution:(13)$$f^{ME}\left(w|m\right)=\frac{e^{S(f(a|w,m))}e^{-\beta \;tanh(\frac{w-\mu_{m}}{T_{m}})w}}{\int e^{S(f(a|w,m))}e^{-\beta \;tanh(\frac{w-\mu_{m}}{T_{m}})w}\;dw}$$

Equation (Equation 13) gives the form of each group-specific marginal wage distributions in the QRSE model. We can obtain a range of interesting conditional distributions from the rules of probability. In particular, we can find the joint distribution of wages and ethnicity as f(w,m)=f(w|m)f(m) (where the f(m) are set equal to the group proportions $\frac{n_{m}}{\sum_{j}n_{j}}$) and then the joint distribution f(w,a,m)=f(w,m)f(a|w,m).

The β parameter is the Lagrangian multiplier associated with the impact of competition on segmental wage rates. This parameter represents the entropy benefits of relaxing the constraint on the differential in the expected wage of entrants and leavers (Equation 5). In other words, β measures how much more uncertain (in bits) an observer would be about the wage outcomes of a worker if the effectiveness of competition was relaxed. In the asymmetric case there are two β parameters, one from the constraint on the differential between the expected wage of leavers from the mean ($\beta_{exit}$) and similarly for entrants ($\beta_{enter}$).

The MaxEnt distribution leaves $\theta =(T_{m},\mu_{m},\beta )$ undetermined. We estimate these parameters by considering (Equation 13) as the kernel to a multinomial likelihood of the observed frequency distributions given θ. Details on the estimation method can be found in the Appendix A. Note that our data are on $\left(w,m\right)$, while the action variable is unobserved. Thus, the MaxEnt model $f^{ME}\left(w,m\right)$ is fitted to the observed frequency distribution f(w,m).

## 4. Data

The data sources for this study are the US Census for 1970–2000 and the American Community Survey, pooled across 2007–2011. For all years, public-use samples were obtained from [31]. These samples are of substantial size, ranging from 1.4 to 5.4 million observations. Samples are restricted to respondents reporting a positive wage and at least some work last year. The sample includes only those living in households according to the 1970 definition; see [31] for details.

Those not currently in dependent employment, in particular self-employed respondents, are excluded from the sample. This restriction is noteworthy, since immigrants differ from native groups in their propensity toward self-employment, with substantial variations among ethnic groups [32,33,34]. In the Smithian view, self-employment ought to participate in the competitive organization of rates of return with forms of dependent employment. However, the appropriate ways of accounting for the different “advantages and disadvantages” ([12] p. 201) of self- as compared to dependent employment are left for future research.

Two observed degrees of freedom are considered in this study, racial/ethnic identifiers and labor incomes. The delineation of ethnic and racial boundaries is a contested and ambiguous task [35]. For the illustrative purposes of this study, we distinguish between Hispanic, Black and white immigrants, and consider their competitive behavior relative to the corresponding native-born groups. We exclude respondents born to American citizens abroad, a group whose immigration experience is sufficiently different to require separate treatment. Ethnic/racial identity is based on the *RACESING* and *HISPAN* variables recorded in [31]. Other ethnic and racial groups are excluded from the sample. There are not enough Black immigrants in the 1970 sample to reliably estimate their marginal wage distributions (see Table A1 in Appendix B). This group is therefore dropped from the analysis as well.

Weekly wages are constructed by dividing total annual pre-tax wage and salary income received as an employee by the number of weeks worked during the previous year. Since the number of weeks worked is given in intervals, I follow [36] in imputing the relevant averages from earlier surveys where detailed information is available. (Similar imputations were made for the weekly hours worked variable in the year 1970.) As discussed below, weekly wages account for differences in unemployment and inactivity spells, which can be substantial. Since women in particular have a higher prevalence of part-time work, weekly wages also tend to accentuate differences in access to full-time employment. (Unfortunately, census data does not allow for distinctions between main and other jobs, and all wages from dependent employment are pooled together.) Finally, the wage distributions f(w|m) for each identity group are estimated using a simple histogram approach with 29 bins of equal size. Within each wage bin, survey weights attached to each respondent are summed up to obtain the estimated frequencies. Observed wages are truncated at the 98th percentile. The resulting histograms of the conditional wage distributions are the basis for the plots and model estimates in this paper. A table of sample sizes and summary statistics can be found in Appendix B.

Wages are deflated using the 1999 Consumer Price Index for All Urban Consumers (CPI-U). This inflation adjustment does not account for cost-of-living differences among individuals within a given census year, such as those which exist between urban and rural areas [37]. Some of the wage inequality observed in the data might reflect spatial differences in the cost of living, without necessarily producing differences in welfare. Between-group differences in wages might similarly in part reflect spatial clustering, a well-observed phenomenon for immigrant [38] and other populations. This paper is silent on the welfare implications of the observed wage differences.

## 5. Labor Competition among Broad Racial and Ethnic Groups in the US

The simple version of the Smithian labor competition model involves a single, typical agent responding to wage differentials and causing feedback effects through her mobility behavior. Even without further disaggregation by social identity and economic characteristics of the workers involved, we can begin to approximate the observed wage distribution for the population of workers in the US. We first review the results of fitting three different versions of the QRSE model class to the population data, and then move forward with our preferred specification to an analysis of different ethnic and nativity groups.

As illustrated in Figure 1 for the year 2010, the simple symmetric QRSE model accounts for the supra-modal range of wages reasonably well. The model fails decisively, and not very surprisingly, for low wages. In particular, the model predicts a large share of negative wages, an economically not very meaningful result.

One approach for dealing with the non-negativity of wages involves simple truncation. Here, the probability mass is set to 0 for all negative wages, and the distribution is renormalized to sum to 1 over the remaining range. As can be seen from Figure 1, this slight modification leads to a dramatic improvement in model fit, both below and above the modal wage. Unfortunately, the parameters of the truncated QRSE do not have an obvious relationship to the parameters of the underlying QRSE distribution (see also, [39]).

The last parametrization we explore allows for the asymmetric impact of competition below and above the mean. As shown in Figure 1, this model similarly provides a much improved fit. Unlike the truncated QRSE model however, the asymmetric version gives an economic account for the precipitous decrease in observed wage frequencies in the infra-modal wage range, compared to the much more gentle decline above the modal wage. Of course, the asymmetric beta QRSE model could similarly be truncated at 0. This is not done here to highlight how closely the untruncated model fits the data. In the version of the model explored here, the asymmetry is due to the differential *impact* of competition on segmental wages, but alternative specifications (perhaps involving asymmetric *responsiveness* in competitive behavior) could be explored in future work.

The informational distinguishability (*ID*) index proposed by [40] offers a useful quantification of the goodness-of-fit between the observed and predicted distributions (Figure 2). The index is defined as $ID\left(p,q\right)=1-e^{-D_{KL}\left(p||q\right)}$, where $D_{KL}\left(p||q\right)=\sum_{k}p_{k}\cdot log\left(\frac{p_{k}}{q_{k}}\right)$ is the Kullback–Leibler divergence between the predicted and observed distributions. We may think of models as devices to compress, and hence efficiently communicate economic data. From this point of view, the *ID* index represents the share of information left to be learned about wage outcomes beyond the Smithian theory of competition.

The estimated *ID* measures confirm the substantial improvement in fit for the truncated and asymmetric models relative to the simple symmetric case. Despite their parsimonious setup, these two model specifications fit the wage distributions remarkably well, generally accounting for 95% or more of the information contained in the group-specific wage distributions. Two exceptions are Hispanic and white immigrants in 2000 and 2010, for which the model strains to fit the data. This is suggestive of features of the data that have not yet been accounted for in the chosen constraints, as will be discussed in the section on future research. The model also fails to converge for white immigrant women in 2010. This category has therefore been dropped from the analysis below. Despite the great simplicity and parsimony of the truncated symmetric QRSE distribution, it is preferable to retain the economic meaning encoded in the MaxEnt constraints. Unlike the truncated model, the asymmetric specification provides an economic intuition for the rapid decline in the observed wage frequencies close to 0. We therefore explore below the results for the asymmetric QRSE model, while leaving to future work the implications of alternative specifications.

### Asymmetric Labor Market Competition between Immigrant and Native Workers

The asymmetric Smithian labor market competition model can be fruitfully applied to the case of multiple groups competing in the same labor market, but with group-specific behavioral parameters. Figure 3 shows the observed distribution of wages for men of different race and nativity groups for each census year (results for women are shown in Figure A1 in Appendix B). Overlaid are the fitted marginal wage distributions and the conditional wage distributions of entrants and leavers.

The model appears successful at fitting most distributions well (Figure 3). For groups characterized by extensive skew—such as white immigrants in recent years—the fit is less adequate. As argued in the discussion section, this can give us indications of missing constraints governing the allocation of wages. A closer look at the conditional wage distributions reveals the significance of entrants in shaping the observed marginal wage distributions for all groups, indicating a preponderance of “entry” in the inferred behaviors. This intuition is confirmed by analyzing the estimated quantal response curves representing the responsiveness to job opportunities for each ethnic group (Figure 4). Overall, workers processes wage information around rather low wage levels very quickly, shifting into and out of jobs as wage conditions change. With higher wages, there is no behavioral change because any job would very likely be accepted. Significant differences between genders are also apparent, with women generally being more responsive to differentials around lower wage levels.

Figure 5 shows the trends over time for the estimated QRSE parameters in each group. The estimates reveal a shifting stratification among both natives and immigrants, and substantial differences between men and women. Whites generally have the highest behavioral temperatures ($T_{m}$), while Hispanic and Black immigrants have the lowest. Native-born Black and Hispanic workers tend to be fairly similar in terms of both behavioral temperatures and reference wages. The most notable feature, however, is the divergent evolution of behavioral temperatures between Hispanic and white immigrants. This divergence coincides with the significant recomposition of the immigrant population in the US, away from European towards Latin American and Asian countries of origin, so that the behavioral temperature of the immigrant population as a whole has decreased substantially over the period from 1970–1990.

In order to appreciate the significance of these behavioral predictions, it is useful to identify what might be called the “competitive reach” of each social group. The competitive threat each group exerts may be summarized by the wage level at which the vast majority of workers would enter a labor market segment. In 2010 for example, 95% of Hispanic immigrant males would accept a weekly wage above roughly 615 USD (in terms of 1999 purchasing power), down from 1040 USD in 1970. The corresponding threshold of white immigrant males was 2710 USD (up from 1670). How do these values compare to the native male population? For Black natives, the 95% threshold in 2010 was 722 USD (down from 1058 USD in 1970), while for white natives the value remained roughly constant at 1660 USD. Native-born Hispanic workers saw a small decrease from 1230 USD to 1115 USD.

These results suggest that among the groups considered here, white native workers are increasingly shielded from competition for higher wage labor market segments, except from white immigrant workers. Most of the competitive behavior of Black and Hispanic immigrants is confined to an increasingly narrow band around relatively low wages, although there is a substantial overlap with native-born Hispanic and Black workers, for whom they remain significant potential competitors. Black workers have seen some convergence towards the values of Hispanics, but there has been almost no improvement in the position relative to whites, certainly in the decades after 1980.

For women, the changes over the same period are less dramatic, with a smaller but still pronounced relative decline for Hispanic immigrants. Group differences overall are less pronounced, with all groups competing around lower wage levels than men. The relative position of the different social groups is mostly similar to men (Figure 4).

Returning to the US labor markets as a whole, we find interesting variations the competitive feedback parameters $\beta_{k}$ (Figure 6). The value of β indicates the contribution of the competitive feedback effect to the organization of wage outcomes (i.e., to the reduction of entropy), so it can be understood as a measure of the impact of competition.

A number of features stand out from Figure 6. First, competition has a greater impact on wages for women compared to men throughout the period of analysis. The effect of leaving is substantially *greater* than the effect of entering, with $\beta_{exit}$ larger than $\beta_{enter}$ throughout. Finally, we find a decline in the competitive impact of entry over the last decades of the 20th century, but little change in the wage impact of labor segment exit except for males in the 1980s. The cumulative effect of workers entering labor market segments therefore is increasingly less effective at competing wages down. Exit on the other hand has about the same effect on organizing wage outcomes as it did at the beginning of the period. At the same time therefore as group-specific measures of job mobility have diverged between social groups, the labor market as a whole has become less competitive. This is how the Smithian model class interprets the rise of wage inequality in the US over the past decades.

## 6. Discussion

The application of the Smithian class of labor competition models to immigrant and native workers in the United States yields a series of interesting results that resonate with other findings in the literature on migration and labor markets.

The competition of immigrant and native workers has been a perennial theme in the economic literature (e.g., [5,36,41,42]). Generally, the focus is on immigration as a supply shock in labor markets and its impact on average wages for different categories of workers. Underlying this literature is a model that interprets observed average wages for different types of workers as the attained equilibrium between supply and demand. Interest centers primarily on whether changes in the supply of a particular type of labor are detectable in these market clearing wages, and whether there are disemployment effects for native workers.

This paper argues for a less mechanistic interpretation of the competition between and among immigrant and native workers. Labor markets are sub-divided into a myriad of niches or segments which allow for varying degrees of mobility. The empirical literature on segmented labor markets either specifies the boundaries of the segments by occupation or industry, or uses switching models [43] to determine membership in segments. Our model is more flexible than these approaches by giving up on the precise specification of labor market segments in favor of a simple measure of segmentation for the market as a whole.

Natives and immigrants may differ both in their evaluation of job opportunities in different labor market segments, and their ability to realize those opportunities. This type of segmentation has been documented in the literature, although its measurement has previously been dependent on an ex-ante delineation of labor market segments [44]. The model presented in this paper can account for this type of heterogeneity, which may reflect institutional [45] and informational barriers [46] as much as employment discrimination [47]. While immigrant and native workers compete in the same labor market, there may be substantial differences in their access and willingness to enter different labor market *segments*.

Indeed, the US labor market is characterized by significant cleavages along race and ethnic lines. Field experiments and audit studies reveal highly persistent discriminatory practices in hiring decisions [48,49]. Perhaps even more important in shaping divergent labor market outcomes between racial and ethnic groups is the cumulative impact of discrimination throughout the life course (e.g., [50,51,52]), and inter-generationally [53]. Immigrants are occupying an increasingly disadvantageous range of positions within this heavily segmented labor market.

The lower behavioral temperature of Hispanic immigrants compared to whites emerges as a robust finding. One possible explanation concerns the selectivity of migration, which would tend to favor workers that are particularly attuned to differences in conditions of employment. Immigrants likely continue to evaluate different job prospects with reference to conditions in their home country, even after having settled in their destination [54,55]. The estimated entry and exit behavior of different immigrant groups provide some support to this notion.

The Smithian class of labor market competition models is arguably consistent with a range of more specific models proposed in the literature. In particular, the framework is agnostic with respect to the factors responsible for generating labor market segmentation. Jobs might differ in terms of the surplus available for bargaining, including due to imperfect competition in product markets and differential exposure to international competition [56].

The heterogeneous agent version of the Smithian class of models allows to broaden the analysis of labor market segmentation to include inter-group dynamics, another aspect of labor competition that has received continued attention in the literature. Large differences in competitive behavior between groups, as can be found between foreign-born and native workers, may for example reflect efforts of native workers to insulate themselves from the competition of immigrants, along the lines of split-labor market theory [1]. The same finding would also be consistent with the strategic use of social identity by employers to segment the workforce in the service of labor control and discipline [2]. Both types of processes, exclusion by native workers and strategic segmentation by employers, are likely to be accompanied by self-reinforcing feedback loops through the use of social networks in hiring and job search [57].

In addition to the differences between the different ethnic groups, the QRSE analysis has also uncovered perhaps even greater differences between men and women. While the approach taken here precludes an analysis of the competitive interaction *between* the genders, it offers an interesting interpretation of the well-known gender differences in wage outcomes [58]. The Smithian model suggests that competition has a greater impact on wages among women compared to men, and women of all ethnic groups are more responsive to wage differentials around lower wage levels. This finding likely reflects differences in labor force participation, with women facing greater unpaid care burdens and discriminatory hiring practices [59].

The period of analysis in this paper covers important institutional changes in the political economy of the United States. As mentioned above, the evolution of the QRSE model parameters can be interpreted as reflecting some of these institutional changes, while others are more challenging to accommodate within the framework. For example, the value of the federal minimum wage has declined in real terms since its peak in 1968 [60]. It is plausible that the decline in this wage floor has rendered workers more vulnerable and eager to accept job prospects that would have been rejected in earlier periods. In addition to changes in the support of the distribution, such effects would perhaps be reflected in declining behavioral temperatures.

Other important determinants of workers’ bargaining power, such as the pro or anti-labor stances of different governments, are hard to incorporate into the model. It is worth noting that the period between 1970–2010 was marked by the continued decline of labor union density among American workers [61]. At least in the US, the inequality-reducing effects of unions among those covered by collective bargaining contracts appear to have been stronger than any potential inequality-enhancing effects between covered and non-covered workers [61,62]. If so, the Smithian model is liable to incorrectly interpret low wage inequality to be indicative of more intense competition among workers, rather than the result of collective bargaining. This illustrates the need to complement any QRSE analysis with a discussion of the political-economic context.

The QRSE labor competition model in this study also contributes to the literature on wage distributions. Noticing their positively-skewed and fat-tailed appearance, some authors have characterized the wage distribution as an exponential (e.g., [63,64,65]). The main benefit of exponential distributions is their extreme parsimony, being defined by only one free parameter. This parsimony comes at a price however, particularly if we are interested in a more finely tuned analysis of the bottom rung of the labor market. The exponential distribution predicts the highest frequency for wages just above 0, or any other minimum wage level one cares to specify. While this dovetails with classical political economy ideas that wages are regulated by historically-specific levels of subsistence [66], there is a non-negligible infra-modal part in observed wage distributions for which the exponential model fails. The QRSE model on the other hand accounts for this regularity by allowing for out-mobility of workers from low-wage labor market segments.

### Limitations and Avenues for Future Research

Applications of statistical equilibrium methods to socioeconomic systems, like all scientific accounts of the world, have to navigate the tension between parsimony of description and the richness of social life. Adam Smith’s analytical emphasis has been on the aggregate results of uncoordinated interactions over long periods of time, so that most distinctions between individuals are irrelevant. (This analytical stance is evident, for example, in Smith’s argument that observed differences in talent (and hence productivity) are primarily the result of “habit, custom and education” rather than “nature” ([12] p. 120).) This article has explicitly introduced distinctions between *some* social groups in the competitive process. However, there are a range of other approaches to the problem of heterogeneity that could be envisaged. It may be possible to allow for more flexible distinctions between social groups, for example using a multi-level parametrization.

One important avenue for future research is the explicit inclusion of employers into the model. The Smithian statistical equilibrium model interprets the observed wage distribution as the snapshot of a dynamical process of adaptation. In the classical view of political economy, labor is able—over a sufficiently long time-span—to adapt quantitatively and qualitatively to the requirements of capital accumulation. Mechanisms such as migration, secondary-earner effects, and educational investments in response to wage differentials are examples of such adaptive processes. Since the labor demand side is only implicit in this model, it is not suitable for assessing, e.g., the surplus available from exchange.

An important limitation of this study is the neglect of changes in the relative rates of labor force participation between groups. For simplicity, the proportion of workers from each group was constrained to equal the observed proportions. This omission is particularly relevant since there are well-known systematic differences in the elasticities of employment for different groups. The role of immigrant workers in regulating the labor supply, for example, has been widely recognized in the literature [67,68,69]. Future research might endogenize both the proportion of agents of each type, and the total number of workers in the system, as part of the competitive process itself.

While the proposed specifications of the QRSE model were able to account for the observed wage distributions quite well in most cases, there were several groups for which the fit was less adequate. The MaxEnt heuristic indicates in such cases that there are other constraints governing the allocation of wages that are not currently included. One important limitation arises from the limited ability of the proposed models to account for skew. Reference [6] introduce skew by adding a constraint on the mean wage. Subsequent work showed that this constraint is equivalent to allowing for disappointed expectations, such that the reference value μ deviates from the realized first moment [9]. Following a similar approach for the wage data leads to the economically not very meaningful result that “entry” is the near-universal response at any positive wage level. Nevertheless, it would be desirable to find ways of relaxing the assumption of realized expectations. This may require a broader interpretation of the advantages and disadvantages of different lines of employment.

## 7. Conclusions

The quantal response statistical equilibrium (QRSE) model, developed in [6], suggests a novel way of analyzing labor market competition between and among different social groups. Grounded in basic economic theory about the unintended consequences of individual behavior, the model proposes a parsimonious informational account of wage inequalities. With only two parameters per group and one or two market-wide parameters we can capture a large share of the information present in wage distributions.

Furthermore, the QRSE model allows us to extract unobserved behavioral information from the observed wage distributions. Despite not having direct observations on behaviors, we can infer both a measure of the responsiveness of workers to wage differentials and a measure of the cumulative impact of individuals’ actions on wages in different submarkets. These measures also give an indication of the degree of segmentation in a labor market, without relying on ex-ante specification of the segments. This application illustrates the usefulness of the MaxEnt principle for socioeconomic analysis in an incomplete data context.

Above, we have investigated what this model reveals about the structure of labor market segmentation by ethnic group in the US. It shows heterogeneity in the job mobility behaviors by gender, and increasing divergence of whites and Hispanic immigrants over the past decades. Native-born African-Americans and Hispanic workers show a very similar competitive reach in terms of the range of wages for which they are particularly sensitive to change. At the level of the US labor market as a whole, our model indicates a substantial decrease in the degree of competitive organization of wages.

## Figures and Tables

**Figure 1 entropy-22-00742-f001:**
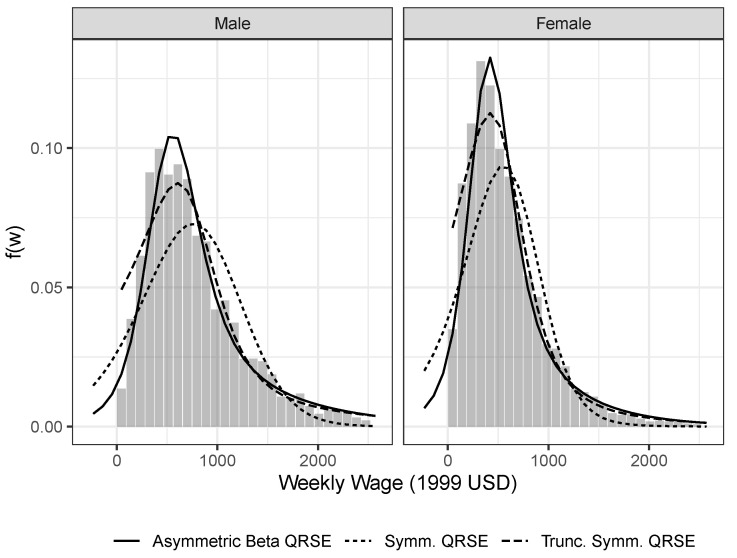
Histogram of marginal wage distributions for 2010 ACS data by sex. Overlaid three different specifications of the QRSE model fitted to the data.

**Figure 2 entropy-22-00742-f002:**
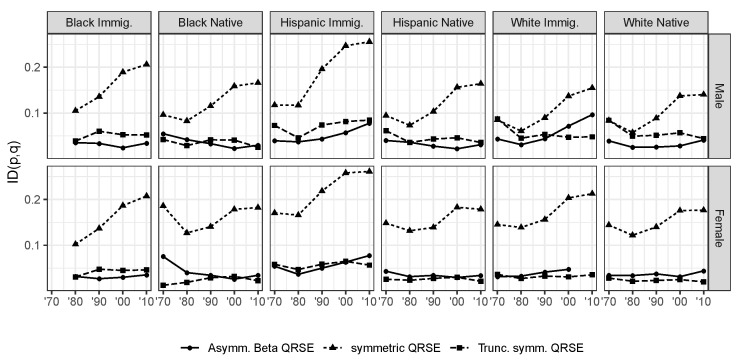
Soofi’s informational distinguishability/goodness of fit measure for the three QRSE model speficiations, by ethnicity. 1970–2010 Census and ACS data.

**Figure 3 entropy-22-00742-f003:**
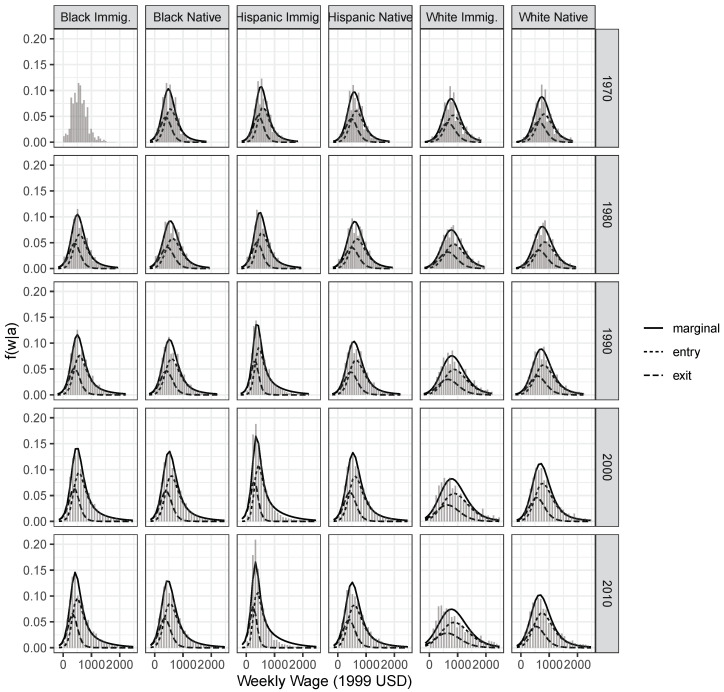
Histogram of observed wage distributions for men, and overlaid fitted marginal (solid) and conditional (dashed) wage distributions. Estimated jointly for all ethnicity groups using the asymmetric Beta QRSE model. 1970–2010 Census and ACS data.

**Figure 4 entropy-22-00742-f004:**
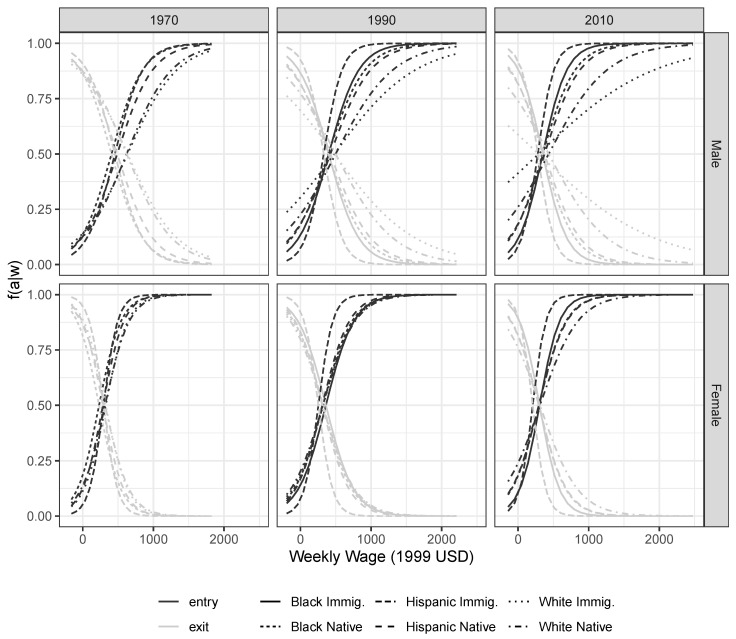
Predicted behavioral responses by race/ethnicity and gender. Selected years, Census and ACS data.

**Figure 5 entropy-22-00742-f005:**
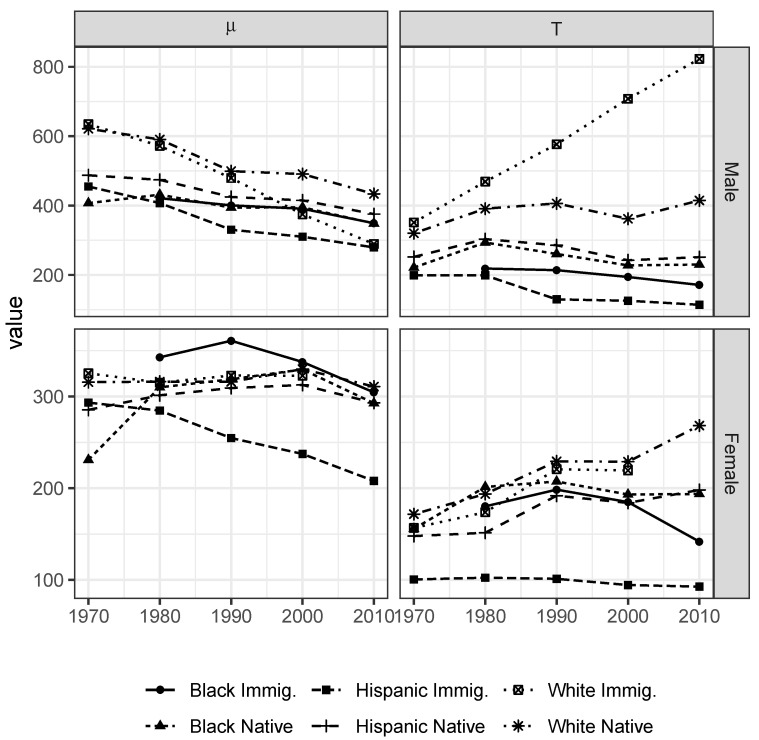
Quantal response reference wages ($\mu_{m}$) and behavioral temperatures ($T_{m}$) by race/ethnicity and gender. Estimated from the asymmetric QRSE model. 1970–2010 Census and ACS data.

**Figure 6 entropy-22-00742-f006:**
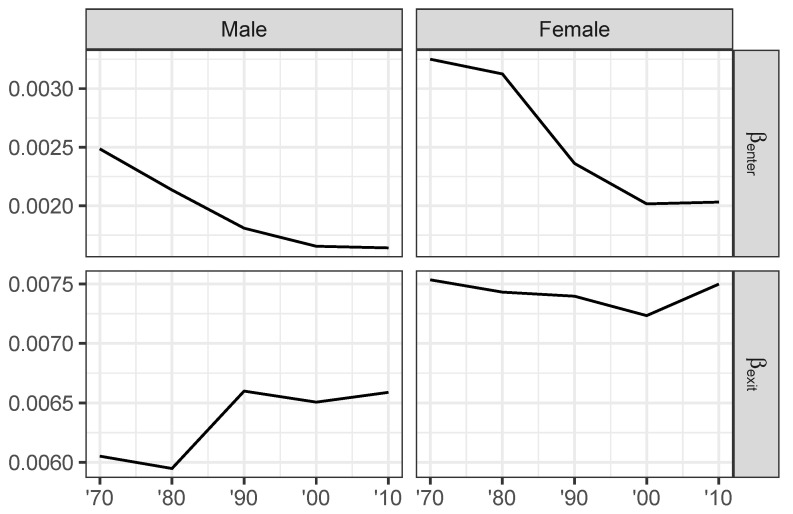
Estimated competitive feedback parameters for the impact of exit ($\beta_{exit}$) and entry ($\beta_{enter}$) by gender. 1970–2010 Census and ACS data.

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
