# Peer review of "Labor Market Segmentation and Immigrant Competition: A Quantal Response Statistical Equilibrium Analysis"

_entropy, 2020, doi:10.3390/e22070742_

Round 1

Reviewer 1 Report

This is a well-written article on analyzing labor markets using Maximum Entropy models.  I am not qualified to comment on the economic aspects of this article, so I will comment only on the methodology.

My primary concern-- and it is a big one that prevents me from recommending acceptance-- is that flawed previous research (not by the author) is used to make constraints for the author's MaxEnt model.  It could be that there are just minor typos in Eq. 2, but I found the following to be troubling:

  • Eq. 2 is not MaxEnt, but is closer to MinEnt solutions, which are notoriously hard to find and almost never can be obtained in closed-form
  • The first constraint is actually not a constraint at all \sum_a f(a|w) (w-\mu) = w-\mu because \sum_a f(a|w)=1, as given in the second constraint. So there's no way that Eq. 2 holds.

In general, if you have both observed and unboserved variables and make MaxEnt distributions, finding the MaxEnt distribution (with Lagrange multipliers solved for) is hard and computationally expensive, e.g. Boltzmann machines.  In other words, I'm very skeptical of Ref. 6.  This is certainly not the author's fault, but I would ask the author to check that they have correctly copied the optimization problem from Ref. 6 and, if they have, I would ask them to check whether or not Ref. 6 actually has a flaw as I have outlined in the two bullet points above.

Usually, misspecifications of previous research by others or flaws in previous research by others aren't enough to cause a "major revision" recommendation, but the author uses results of Ref. 6 to choose his constraints.  As such, I need to see evidence that Ref. 6 is actually correct before I recommend complete acceptance of the manuscript.

Reviewer 2 Report

The paper analyzes wage differentials between native and immigrant workers in the US using Census data and applying a Quantile Response Statistical Equilibrium Analysis.

The paper is interesting, well structured and well written, but there are several aspects that deserve the author's attention before it can be potentially considered for publication in Entropy

Major issues

1) About the design of the research 

1.1. In my view, it is absolutely necessary to consider the gender dimension in the empirical analysis. Most labour economists analyse the US labour market focusing on men working full-time only. This is not my recommendation but it is a clear sign that gender segmentation is present in the US market and should be considered in the analysys 

1.2 Immigrants are a very heterogenous group in terms of origins (and skills) but also in the characteristics that favour a better integration in the labour market (age, language knowledge, … ), but also race and ethnicity. I suggest dividing the sample in at least two groups: those competing with whites and those competing with black people. In fact, you are aware that compositional effects are relevant as shown in line 302.

1.3. The paper uses weekly wages. Do they refer to the main job? It is very important to state what has been your treatment of multiple job holders. Focusing on weekly wages instead of anual wages (or hourly wages) is also a very important decision as unemployment and inactivity spells would also be very different for each group of workers and clearly affect the results of the analysis.

1.4. Related to the previous point, do you include self-employed workers in your analysis? Again group differences are relevant 

1.5. Line 212. assuming that labor force participation is exogenous is a very important limitation as different groups can react very differently to Business cycle conditions. Please acknowledge it as a limitation in the concluding section.

1.6. Have you used Census sampling weights in your analysis?

1.7. line 246 - 29 bins are used to plot the data only or also for computations? 

1.8. Do CPI deflators used in your analysis consider regional differences in the cost-of-living? Residential segregation in some states and in metropolitan/non-metropolitan areas could also explain a relevant part of nominal wage differences.

1.9. In my view the selection of the asymmetric QRSE model is not justified in view of the results. The model that makes more sense in economic terms (no 0/negative wages) but also according to the results shown in figure 2 is the truncated QRSE. Why not using this model for the analysis?

2. Other aspects 

2.1. Line 12 - Wage determination is not only affected by worker competition. Labour market institutions and regulations such as mínimum wage or collective bargaining are relevant in most labour markets to shape wage distributions. 

2.2. Line 13 - I think you could refer to individuals and not households across the paper. While some joint decisions related to labour supply are taken at the household level, your analysis omits this dimension and focuses on individual decisions.

2.3. Line 83 - I do not agree that Censuses are the most extensive data sources on labour market outcomes. Nowadays, most empirical papers in Labour Economics are based on longitudinal matched employer-employee datasets coming from surveys or from administrative registers.  

2.4. In fact, I am a bit surprised that there is no role for firms (labour demand) in your analysis. Aren't you assuming that there is a fixed amount of labour available that cannot vary in the short term?

2.5. Line 382 replace income by wages

2.6. Line 409 Wage assimilation is a dynamic process taking place at the individual dimension. I think that you cannot infere about assimilation when looking at results for the average immigrant and native. Cohort quality changes over time could explain the result and not a better integration of immigrants in the labour market.

2.7. I recognise that I did not understand the issue of "incomplete data" mentioned in section 2. Is this a relevant point for the analysis? I think it is not. I suggest dropping it or elaboraring more this idea in the specific empirical application.

Minor issues:

1) there are a very few typos that require correction (i.e., line 180 infintitely)

2) Figure 2 should be mentioned in lines 271-272.

3) A supplementary table could be provide with descriptive statistics of wages by group and year.

I hope that you will find these comments appropriate and useful to revise your paper. 

Round 2

Reviewer 1 Report

I am now happy with the manuscript.

Reviewer 2 Report

I am satisfied by the way the authors have considered my comments on the previous version. I now recommend publication in Entropy. Congratulations!